# ‘Making the System Work’: A Multi-Site Qualitative Study of Dietitians’ Use of iEMR to Support Nutrition Care Transitions for Older Adults with Malnutrition

**DOI:** 10.3390/healthcare13172227

**Published:** 2025-09-05

**Authors:** Kristin Gomes, Shelley Roberts, Ben Desbrow, Jack Bell

**Affiliations:** 1School of Health Sciences and Social Work, Griffith University, Gold Coast Campus, Southport, QLD 4222, Australia; s.roberts@griffith.edu.au (S.R.); b.desbrow@griffith.edu.au (B.D.); jack.bell@health.qld.gov.au (J.B.); 2Allied Health Research, Gold Coast Hospital and Health Service, 1 Hospital Blvd., Southport, QLD 4219, Australia; 3Allied Health Research Collaborative, The Prince Charles Hospital, 627 Rode Road, Chermside, QLD 4032, Australia

**Keywords:** electronic medical records, discharge planning, malnutrition, older adults, care transitions, care continuity

## Abstract

Background: Older adults with malnutrition (≥65 years) require coordinated nutrition care during hospital-to-home transitions. A key purpose of integrated electronic medical record (iEMR) systems is to support clinicians in ensuring continuity of care across settings, yet little is known about their use in nutrition care discharge practices. This study explored how clinical dietitians use the iEMR to support nutrition care discharge practices for older adults with malnutrition and identified opportunities for optimisation to enhance care continuity. Methods: Semi-structured interviews were conducted with 16 clinical dietitians (11 frontline clinicians, 5 senior leaders) from 10 public hospitals across Queensland, Australia. Analysis combined deductive coding using the Consolidated Framework for Implementation Research 2.0 with inductive thematic analysis to identify system-level, organisational and behavioural influences on iEMR use and optimisation opportunities. Results: Four themes and ten subthemes were identified. System fragmentation, policy constraints and documentation burden limited dietitians’ ability to coordinate discharge care. Workarounds were common and reflected both practical adaptation and conditional trust in iEMR. Discharge practices were also shaped by local culture, professional norms and variable expectations for iEMR use. Despite these constraints, participants expressed aspirations for an optimised iEMR with embedded referral tools, real-time alerts and analytics to support improved service delivery. Conclusions: This study identified key factors influencing iEMR use by clinical dietitians to support nutrition care transitions for older adults with malnutrition. While current systems present significant challenges, optimising iEMR alongside organisational and policy enablers holds potential to strengthen nutrition care discharge practices and care continuity.

## 1. Introduction

The digitalisation of healthcare is reshaping delivery of complex nutrition care across settings [1]. For hospitalised older adults (aged ≥ 65 years) with malnutrition, who require multidisciplinary care to recover, this holds particular significance. Malnutrition affects up to 50% of hospitalised older adults globally [2], including 30–40% in Australia [3]. Its consequences often extend beyond the inpatient stay, contributing to functional decline, increased healthcare utilisation, reduced quality of life and substantial healthcare costs [2,4,5]. Ensuring continuity of nutrition care after discharge is critical to supporting recovery and preventing readmissions [2,6,7,8], especially as this transition remains a period of heightened vulnerability [7,9,10].

Research indicates that post-discharge nutrition support for older adults with malnutrition improves dietary intake, weight status and quality of life [11,12]. However, few studies have examined how nutrition care is coordinated as older patients transition from hospital to home. Evidence suggests that nutrition care discharge plans are inconsistently provided, poorly adhered to and often fail to ensure adequate post-discharge support [13,14,15,16,17,18]. In most hospitals, dietitians are responsible for facilitating the nutrition-related aspects of discharge planning and care coordination [19,20]. In Australia, community dietetic services are available through public health, specialist outpatient clinics, and private providers; however, access is criteria-based and variable depending on funding and capacity [21]. As such, hospital-based dietitians retain primary responsibility for nutrition care planning and coordination prior to discharge [21]. Interviews with clinical dietitians employed within a tertiary hospital in Australia recently indicated their discharge practices evolve through experience and must accommodate local team norms, workarounds and adaptations to system constraints [21]. These findings underscore that continuity of care is not solely dependent on clinician intent, but on the systems that enable or constrain care delivery.

Integrated electronic medical record (iEMR) systems are widely implemented across hospital settings to support real-time documentation, interdisciplinary communication and clinical workflows [22,23] and are recognised in national digital health strategies, including Australia’s National Digital Health Strategy [24], as key enablers of safer care transitions [1,25]. While allied health professionals generally report positive attitudes towards iEMR systems [26,27], they have raised concerns about system complexity, inadequate training and documentation burden [28,29]. Indeed, a recent chart audit exploring nutrition care discharge practices by dietitians revealed wide variation in both approach and documentation, with nutrition care recommendations documented in the Electronic Discharge Summary (EDS) for only 10% of cases, despite routine iEMR use [18]. This result parallels evidence across allied health professionals, which indicates most users engage with iEMR for basic documentation and retrieval functions, while underutilising advanced tools (e.g., integrated care planning) [30]. Concerns over suboptimal iEMR use has resulted in leading nutrition, dietetics and clinical information management agencies to issue recommendations for optimising electronic health records to better support nutrition care, including structured documentation workflows and improved care plan visibility during transitions [31]. Beyond nutrition-specific guidance, large-scale iEMR implementations have also demonstrated benefits such as improved documentation quality, enhanced care coordination and reductions in adverse events [32,33]. To date, it remains unclear whether current iEMR systems enable dietitians to optimally plan and coordinate nutrition care at discharge to ensure continuity of care for older adults with malnutrition.

Understanding dietitians’ engagement with iEMR systems is essential to identifying optimisation opportunities and strengthening digital workflows that support continuity of care. This study explores clinical dietitians’ perceptions of how iEMR supports discharge planning and care coordination for older adults with malnutrition transitioning from hospital to home and identify opportunities to optimise its use to strengthen care transitions and nutrition care continuity.

## 2. Materials and Methods

### 2.1. Study Design

This qualitative descriptive study used semi-structured interviews and is reported as per the Consolidated Criteria for Reporting Qualitative Research (COREQ) [34]. It was informed by a critical realist ontology and a pragmatic, interpretivist-informed epistemology, acknowledging that healthcare structures like the integrated electronic medical record (iEMR) exist independently but are experienced and enacted differently across contexts. This orientation supported an exploration of current iEMR use and perceived optimisation opportunities. Ethical approval was granted by the Gold Coast Hospital and Health Services Human Research Ethics Committee (LNR/2023/QGC/103880; approved 13 December 2023) and the Griffith University Research Ethics Committee (reference number: 2023/968; approved 21 December 2023). Site-specific authorisations were granted by participating Hospital and Health Services. All participant information was treated confidentially, with interview transcripts deidentified prior to analysis and stored securely in line with ethics approvals.

### 2.2. Setting

This study was conducted within public hospitals in Queensland, Australia in the advanced implementation phase of the iEMR system. Twelve sites across six Hospital and Health Services (HHSs) were selected based on access to iEMR functionalities relevant to discharge planning and care coordination. All sites operated under Queensland Health’s tiered capacity escalation system, which embeds early discharge planning into routine workflows [35]. These system capabilities were considered relevant to examining how dietitians document and coordinate care during transitions [36].

### 2.3. Participants

Participants included clinical dietitians working in adult acute medical or surgical wards, responsible for inpatient nutrition care, discharge planning and post-discharge care coordination. Purposive sampling ensured variation in service contexts and site representation. Email invitations with study details, participation requirements, the Participant Information and Consent Form (PICF), and researcher contacts, were distributed via local site contacts. Written consent was obtained electronically via signed PICF or email confirmation, with verbal confirmation at the start of each interview. Participation was voluntary and no financial compensation was provided.

### 2.4. Data Collection

All interviews were conducted by the lead researcher (KG), a female Accredited Practising Dietitian and PhD candidate with a Bachelor of Nutrition and Dietetics (Honours) and a Master of International Public Health. KG has professional experience in clinical dietetics and public health, including the design, delivery and evaluation of health programs. Her qualitative research skills were developed through postgraduate training, honours research, and experience conducting interviews and focus groups in research and professional public health settings. For this study, KG received methodological guidance from senior research team members (SR, JB, and BD). While most participants had no prior relationship with KG, a small number were familiar with her through previous clinical roles within Queensland Health.

Interviews were scheduled at participants’ convenience and conducted using a semi-structured interview guide (Appendix A). The interview guide was developed with reference to the Academy of Nutrition and Dietetics’ Nutrition Care Process (NCP) model [37]. The guide explored how dietitians deliver and coordinate nutrition care during hospital-to-home transitions and was reviewed by the research team to ensure relevance and alignment with key components of nutrition care practice. Participants provided demographic information (clinical experience, site tenure, inpatient focus, clinical specialty). KG used a conversational interviewing style directed by the guide, incorporating prompts and follow-up questions to explore emerging ideas.

Interviews were conducted, recorded, and transcribed using Microsoft Teams. Transcripts were reviewed against audio by KG, manually corrected for accuracy and deidentified prior to analysis. Data collection continued until no new concepts emerged. Repeat interviews were not required. Participants were given the opportunity to review and amend their transcript prior to analysis.

### 2.5. Data Analysis

Interview transcripts were analysed using a structured, hybrid qualitative approach combining deductive thematic analysis guided by an implementation framework, followed by inductive thematic analysis and synthesis. This multi-phase strategy enabled a structured yet flexible analysis, identifying key implementation influences while allowing context-specific insights to emerge. Combining deductive and inductive methods is increasingly recognised as a rigorous, pragmatic approach in applied health and implementation science research, particularly for complex system-level processes or digital health innovations [38,39,40,41,42]. This approach aligned with the study’s applied focus and intended audience, supporting the generation of theoretically grounded, practice-relevant findings.

The Consolidated Framework for Implementation Research (CFIR 2.0) was applied post hoc to guide deductive coding [43]. Although the interview guide was not originally framework-informed, CFIR 2.0 provided a comprehensive structure for identifying contextual, organisational and individual-level influences on iEMR use during discharge planning and care coordination. Its application supported consistent coding aligned with the study’s aim to understand how digital systems are used within routine practice and how they might be optimised.

Three transcripts (3 of 16; approximately 20% of the dataset) were independently blind coded by KG and a senior research team member (JB) using the full CFIR 2.0 construct set [43]. Each quote was assigned one primary construct code that best aligned with the data, with a secondary construct code applied when additional CFIR constructs were meaningfully present in the same data segment. This dual coding approach, consistent with CFIR 2.0 guidance [43] and the recently published CFIR User Guide [44], enabled analysis of construct co-occurrence across the dataset. A consensus meeting with KG and JB was conducted to refine coding alignment, establish consistent construct application and contextualise CFIR 2.0 use to the study setting. All sixteen transcripts were then coded by KG using the refined guide, including re-coding of the initial three. Constructs were only applied when meaningfully reflected in participant responses; several constructs did not appear in the dataset, not due to exclusion, but because they did not emerge in the interviews. A total of 1048 quotes were coded across 22 CFIR 2.0 constructs. Coding was completed using Microsoft Excel (Microsoft Corporation, Redmond, WA, USA). This staged approach, initial dual coding followed by single-coder application with consensus refinement, is consistent with qualitative guidance recommending that at least 10% of transcripts be double-coded when full dual coding is not feasible [39,45].

Construct frequency analysis was conducted using Microsoft Power BI (Microsoft Corporation, Redmond, WA, USA), which automated frequency counts and supported visualisation of patterns across constructs, to identify the ten most frequently coded constructs for subsequent thematic analysis. Frequencies were used as a pragmatic proxy to identify dominant areas of focus in participants’ accounts, consistent with CFIR-informed approaches in other qualitative studies [46]. A two-phase inductive process followed:(1)Construct-level thematic analysis: Quotes coded as primary for each of the ten constructs were reviewed to identify between four and five subthemes per construct (45 total). Secondary-coded quotes were reviewed to enhance thematic depth and nuance.(2)Construct co-occurrence analysis and thematic interpretation: Frequency analysis was conducted using the most frequently coded CFIR 2.0 constructs identified during deductive coding to generate all possible construct pairings within this group. This approach searched the full dataset for quotes tagged with both constructs in a given pair, regardless of which was primary or secondary, enabling analysis of how contextual factors intersected across data segments. Quotes coded with both constructs in each pair, regardless of coding order, were analysed inductively. Thematic analysis continued until no new concepts were identified; 21 co-occurrence themes were identified.

An inductive thematic synthesis was then conducted to map co-occurrence themes back to their corresponding individual construct themes, exploring conceptual alignment, divergence and reinforcement. Synthesis summaries were developed from these relationships and grouped into overarching thematic categories. The structure was iteratively refined through review and discussion with the supervisory team to enhance conceptual clarity and finalise the framework of overarching themes and subthemes. All exemplar quotes were tagged to indicate whether they reflected (a) strategic reflections, (b) leadership perspectives, or (c) alignment with Part A (current iEMR use) or Part B (optimisation opportunities) of the study aim. These tags supported thematic synthesis and informed interpretation of findings in relation to the study’s dual aims.

Trustworthiness was supported through multiple strategies. Credibility was strengthened by investigator triangulation, iterative coding, phased team reviews and use of a semi-structured interview guide to standardise data collection. Dependability and confirmability were enhanced through systematic documentation of coding decisions, theme refinements and an audit trail of analytic steps, which also provided a reflexive record of the analytic process. Authenticity was supported through purposive sampling across varied roles, experience levels, and health service contexts. Transferability was promoted through detailed descriptions of setting, participants and analytic processes. Reflexivity was enacted throughout coding and analysis. KG documented and revisited assumptions in field notes, clarified construct application through consensus refinement, engaged in iterative team reviews of draft themes and subthemes and verified interpretations against participant exemplar quotes. KG also reflected on her professional positionality, including the potential influence of prior clinical relationships.

## 3. Results

Results are presented in three parts: (1) participant demographics, (2) deductive coding frequencies and construct co-occurrence patterns (Table 1 and Table 2), and (3) a qualitative thematic synthesis outlining four themes and ten subthemes (Table 3) with selected quotes integrated into the narrative and full-length examples provided in Appendix A.

### 3.1. Demographics

Participant characteristics are summarised in Appendix A and described below. Sixteen interviews were conducted, including 11 frontline clinical dietitians and five senior leaders, spanning ten hospital sites across six Queensland HHSs. The five senior leaders (three directors, one team leader, one research dietitian) are collectively described as ‘senior leaders’ to ensure anonymity. Participants had a median of 6.0 years of clinical experience (IQR 3.0–15.8), with over half (*n* = 9) reporting six years or less. Median tenure at their current site was 3.0 (IQR 1.3–11.5) years. Participants worked across a mix of medical, surgical, or combined caseloads, with four also providing outpatient or telehealth care. Seven frontline dietitians previously worked in paper-based systems, while the remainder had only practiced in digital environments. Sites included metropolitan and regional hospitals with varied local workflows and resources. Interview durations ranged from 16–56 (IQR 34–44) minutes, with a median of 39 min.

### 3.2. Deductive Coding Frequencies and Construct Co-Occurrence Patterns

A total of 1048 quotes were coded across 22 CFIR 2.0 constructs (Table 1). These constructs spanned all five CFIR 2.0 domains, with most codes falling under Inner Setting and Innovation. This distribution highlights the interplay of contextual, innovation-specific, interpersonal and system-level factors shaping iEMR use in practice. The most frequently applied constructs were Compatibility, Partnerships and Connections, and Culture. To identify key influences, we first quantified the frequency with which each CFIR construct appeared in the data (Table 1). These construct frequency patterns then guided the selection of 11 constructs for co-occurrence analysis (Table 2) and subsequent thematic analysis and synthesis (Table 3).

The top eleven constructs in Table 1 were selected for co-occurrence analysis. Innovation Design, ranked just outside the top ten, was included due to its strong conceptual alignment with other innovation-related constructs and its relevance to optimisation opportunities. The most frequent construct pairings linked system compatibility, coordination beyond hospital boundaries and iEMR complexity. The top three pairs were Compatibility and Partnerships and Connections, Compatibility and Innovation Complexity, Partnerships and Connections and Innovation Relative Advantage. These pairs bridged Inner Setting, Outer Setting and Innovation domains, capturing how participants assessed system fit across internal workflows and external coordination processes.

### 3.3. Qualitative Thematic Analysis and Synthesis

Thematic analysis identified four overarching themes and ten subthemes describing dietitians’ perspectives on using iEMR to support discharge planning and care coordination for older adults with malnutrition. These reflect both current practice and future opportunities to optimise iEMR for improved nutrition care continuity. Table 3 summarises each theme and subtheme; expanded interpretations are provided in the narrative below, with full exemplary quotes available in Appendix A.


**Theme 1: Technical system limitations and policy constraints**


System-level design, usability, and interoperability issues, compounded by external policy restrictions, were perceived to undermine the effectiveness of discharge planning and coordination. These constraints disrupted communication workflows, reduced confidence in digital documentation, and limited options for post-discharge nutrition care.


*Subtheme 1.1. Technical fragmentation, poor interoperability and design limitations compromise care*


Participants described iEMR as fragmented and difficult to navigate, with key discharge planning information dispersed across PowerForms, free-text notes and the Enterprise Discharge Summary (EDS). Limited integration contributed to duplication, inefficiencies, and uncertainty about where to document critical information. Usability challenges, including filtered visibility of dietetic notes, rigid EDS workflows and formatting restrictions, further eroded trust in the system’s ability to communicate nutrition care plans. One participant explained that critical handover information often required manual follow-up, with dietitians “double or triple handling everything” to ensure recommendations were received (P05, Dietitian).


*Subtheme 1.2. Policy and program eligibility restrictions limit service access and care coordination*


External funding structures and eligibility criteria tied to national health and social care schemes constrained dietitians’ ability to coordinate post-discharge care. These administrative rules often overrode clinical judgement, requiring referral decisions to be based on eligibility rather than nutritional need. Participants described the processes as opaque and time-consuming, with resulting care gaps contributing to potential readmissions, particularly for nutritionally vulnerable patients who fell outside the funding parameters. As one participant stated, “just hoping that it happens” was sometimes the only option when coordinating with external providers (P07, Dietitian).


**Theme 2: Adaptations and workarounds in daily practice**


In response to the system limitations described above, dietitians’ adaptations reflected both pragmatic workarounds and varying levels of trust in the iEMR system. Concerns about documentation visibility, system reliability and communication gaps shaped confidence in digital processes, particularly during discharge. As a result, clinicians developed parallel communication strategies and practical workarounds to support continuity of care. While these offered short-term solutions, they introduced variability and reinforced reliance on individual initiative over standardised processes.


*Subtheme 2.1. Trust concerns drive reliance on parallel communication methods*


Participants frequently supplemented iEMR documentation with phone calls, emails and verbal confirmations, not to save time, but to ensure their notes were seen and acted upon. Visibility concerns were also reflected in broader doubts about system reliability and value (see Subtheme 4.1), especially when coordinating with external providers. These behaviours stemmed from a deeper lack of trust in the digital handover process. One participant described feeling more confident when “sending an e-mail where you get feedback from that person” rather than relying on the system alone (P04, Dietitian). Although duplicative, this approach provided reassurance that critical discharge information had been received.


*Subtheme 2.2. Documentation and communication workarounds emerge to overcome system functionality limitations*


At both frontline and leadership levels, dietitians described adaptations to bypass iEMR limitations and streamline workflows. Some leaders directed teams to use free-text notes instead of structured templates, which were perceived as cumbersome to navigate. Several frontline clinicians bypassed structured templates entirely, drafted notes in Word, or used printed or faxed summaries when digital workflows were too rigid. One participant avoided typing directly into the templates because “when you edit in the form and save it…it looks terrible”, instead drafting in Word and pasting the note into iEMR to “make it look neater for everyone” (P07, Dietitian). Some preferred direct communication over digital handovers to ensure information transfer. While efficient, such workarounds introduced variability and risked undermining documentation standardisation.


**Theme 3: Organisational culture and norms**


Social, cultural and professional norms shaped how dietitians approached discharge planning and documentation, often beyond the constraints of system design. Inconsistent expectations across teams, legacy documentation habits and varying levels of leadership engagement contributed to divergent iEMR use. Professional identity shaped perceptions of visibility and influence in discharge processes, while interpersonal relationships often took precedence over formal coordination mechanisms. These findings reinforce that system use was shaped more by local culture than technical design.


*Subtheme 3.1. Practice variation is shaped by inconsistent cultural expectations and norms*


Dietitians reported wide variation in expectations around discharge documentation, particularly the use of the EDS, across hospitals, teams and disciplines. Some clinical areas, such as rehabilitation wards, maintained performance-driven mandates for EDS completion, whereas others lacked guidance or accountability. Participants described how legacy norms and siloed team cultures influenced whether tools like the EDS were used. These cultural inconsistencies shaped both how documentation was completed and whether it was prioritised. As one participant noted, “there’s no accountability…no one’s checking completion” even in areas where EDS was technically expected (P01, Dietitian).


*Subtheme 3.2. Dietitians’ evolving role and visibility in discharge planning*


Senior leaders reflected on how cultural norms shaped dietitians’ perceived importance in discharge planning. Dietitians were often viewed as peripheral to discharge decisions, with less influence than other allied health professionals who “have more onus of control over discharge because it’s functional” (P13, Senior Leader). This marginalisation was reinforced by limited policy guidance, weak accountability structures and poor visibility within the iEMR. As one leader noted, others “couldn’t care less if they’d discharged a patient with or without dietetic input” (P12, Senior Leader). These reflections highlighted that visibility was shaped more by organisational culture and professional hierarchies than by individual capability.


*Subtheme 3.3. Relational coordination often supersedes structured workflows*


Discharge coordination frequently occurred through informal relationships rather than structured digital workflows. Dietitians described leveraging rapport with social workers, discharge coordinators and nurses to ensure continuity of care. These strategies helped navigate system limitations and reduced reliance on digital handover tools. Examples included asking a nearby colleague who covered outpatients to “just have a quick look at the patient’s chart” rather than submitting a formal referral (P06, Dietitian), or contacting former colleagues directly by “just call[ing] the admin staff I know there…and send[ing] an e-mail handover to that person” (P11, Dietitian). The effectiveness of these strategies varied by team size, physical proximity, and local culture, illustrating how social capital often replaced digital systems as the main mechanism supporting care transitions.


**Theme 4: Optimising iEMR: future directions and system potential**


While discussing current challenges, participants also envisioned a more intelligent and integrated digital system to support nutrition care transitions. Confidence in iEMR varied, yet dietitians identified opportunities to improve its value through standardised workflows, intuitive interfaces and advanced analytics. These reflections signalled a shift from compensating for system limitations to imagining tools that proactively support coordinated discharge planning and continuity of care.


*Subtheme 4.1. Confidence in system value varies by clinical context and user experience*


Dietitians expressed mixed views about iEMR’s value, influenced by setting, task and whether coordination extended beyond their health service. Many praised its internal efficiency and real-time access to information, but confidence decreased when engaging with external providers or navigating fragmented documentation. Senior leaders were more sceptical, questioning whether iEMR improved upon paper-based processes. One noted it was “hard to say” if progress had been made, given the difficulty locating discharge information across system components (P14, Senior Leader), while another praised improved visibility: “that snapshot of patient data is more easily visible” (P09, Dietitian). These divergent views reflected a nuanced relationship with iEMR, shaped by context, role and discharge scenario.


*Subtheme 4.2. Analytical capabilities hold potential for transforming service delivery*


Participants, especially leaders, described aspirations to use iEMR’s analytical capabilities for service planning, outcome monitoring and patient prioritisation. This subtheme captured forward-looking views about structured data informing system-level decisions. Suggestions included dashboards, automated alerts, and tools to track patient outcomes. One leader described this shift “looking at outcomes more than just activity” (P15, Senior Leader), while another envisioned dashboards as “our data bank for how we prioritise, what the trends look like and how we mobilise to provide care” (P13, Senior Leader). These reflections suggested that consistent documentation practices and leadership engagement could transform service planning and delivery across health services.


*Subtheme 4.3. Integrated and standardised workflows promise more efficient discharge planning*


Dietitians consistently advocated for unified digital platforms and standardised documentation to streamline discharge processes. They described burdens associated with toggling between systems and duplicating information, proposing integrated discharge tools within a single platform. Suggestions included embedding referral functionality, aligning workflows across services and creating a centralised space for end-of-stay documentation. One dietitian imagined “just one sort of area…where you can do your last review note plus the plan post hospital discharge” (P09, Dietitian). These aspirations emphasised the need for simplified workflows that reduce administrative load and support timely, coordinated transitions. One leader noted the need for “far-sighted people to make it standard practice” (P16, Senior Leader), emphasising that system redesign depends on visionary leadership.

Appendix A includes full exemplar quotes supporting each subtheme. Collectively, these four themes and ten subthemes illustrate the interconnected factors shaping dietitians’ iEMR use in discharge planning and care coordination to support continuity of nutrition care for older adults with malnutrition.

## 4. Discussion

This study examined how clinical dietitians use integrated electronic medical record (iEMR) systems to support discharge planning and care coordination for older adults with malnutrition transitioning from hospital to home and identified opportunities for iEMR optimisation to strengthen continuity of nutrition care. Four overarching themes and ten subthemes were identified, illustrating how technical system limitations, adaptive workarounds, organisational culture and optimisation aspirations collectively shaped discharge practices. To our knowledge, this is the first system-level, multisite analysis of clinical dietitians’ engagement with digital systems to support nutrition care transitions, addressing a critical gap in digital health research.

System fragmentation and technical constraints created significant barriers to timely, coordinated discharge planning. Information was difficult to locate, required duplicative entry and lacked visibility across providers during hospital-to-home transitions. These challenges reflect broader digital health concerns, where poor interface design, rigid workflows and fragmented integration undermine efficiency and interdisciplinary communication [29,47,48,49]. Concurrently, some digital health studies highlight iEMRs’ potential to assist care transitions, reporting improvements in documentation access, workflow integration and interdisciplinary communication following implementation [32,33]. In this study, limited interoperability with external providers led dietitians to adopt parallel communication strategies such as email, phone, and fax. These workarounds increased documentation burden, consistent with findings that iEMRs extend documentation time [50]. Policy constraints further complicated discharge planning, as funding mechanisms and eligibility criteria frequently overrode clinical judgement when determining post-discharge care options. These findings suggest that continuity of care depends not only on integrated digital infrastructure but also aligned policy frameworks that support clinical decision-making [51]. Without addressing both, iEMR systems will remain limited tools for coordinated care, reinforcing the workarounds that compromise nutrition care continuity during transitions.

These system and policy constraints also affected how dietitians used the iEMR system and the confidence they placed in it. Many described drafting notes externally, bypassing speciality-specific templates or supplementing digital handovers with phone calls or faxing to ensure information was received. These workarounds reflected uncertainty about whether documentation would be visible or acted upon, particularly across care settings. Prior research confirms these patterns, documenting weaker EMR-supported collaboration between hospital and community settings [52,53], and clinicians’ reliance on informal workarounds to navigate rigid workflows and system fragmentation [49,54]. Trust in iEMR appeared conditional in our study, stronger within the same health service but weaker across organisational boundaries, contributing to communication duplication and redundant documentation. These findings align with evidence that system opacity, fragmented workflows, limited feedback loops and poor interoperability erode trust in digital handover tools [52,53,55,56]. Technical fixes alone cannot resolve these confidence and visibility gaps. Without improved system reliability and clearer visibility across care boundaries, informal workarounds are likely to persist, undermining the standardisation and care continuity that digital systems are intended to support.

While these workarounds were practical responses to system and policy limitations, their persistence was reinforced by social, cultural and professional norms. Discharge planning and documentation patterns reflected not only technical adaptations, but also local culture, legacy practices and perceived role boundaries. Dietitians described inconsistent expectations around tools like the EDS, with use influenced more by team norms and leadership than formal policy. These findings extend earlier qualitative research within the same health system, which found that dietitians’ discharge practices evolved through peer norms, experiential learning and site-specific expectations [21]. That such variation occurred across hospitals within the same system and iEMR rollout phase underscores that local context, not system design alone, shapes how digital tools are used and discharge practices are carried out. Broader digital health literature shows that iEMRs may reinforce professional hierarchies and fragmented information sharing, particularly affecting allied health disciplines like dietetics whose priorities may be less visible in discharge workflows [29,54,57]. While digital readiness remains a barrier [28], dietitians’ reliance on informal networks reflects both adaptations to system constraints and preference for established communication patterns, consistent with previous dietetic-specific findings [58,59]. This reflects broader interprofessional collaboration literature, where relationship-based coordination frequently supplements or replaces formal processes [60]. Discipline-specific documentation reinforced professional boundaries and limited visibility across teams, contributing to communication gaps noted in prior studies [49,54,61]. Improving standardisation in nutrition care transitions therefore requires aligning technical integration with team norms, leadership priorities and workplace culture.

In response to these constraints, participants described a more supportive digital system to strengthen discharge planning. Frontline dietitians called for simplified workflows, embedded referral tools, real-time alerts and a centralised digital space for discharge planning. Senior leaders emphasised iEMR’s analytics potential for workload prioritisation, outcome monitoring and service planning. These complementary perspectives reflected significant unrealised potential. Such aspirations align with broader health informatics literature [26,29,51], as well as Australia’s National Digital Health Strategy and Queensland’s Digital Health 2031 roadmap, which position iEMRs as enablers of safer, coordinated care and improved transitions across settings [24,62]. Translating these aspirations into practice will require improved system design, sustained leadership commitment, end-user engagement and a shift toward data-informed service models [29,54]. Practical strategies include co-designing integrated workflows and referral tools with end users [63], aligning interventions with existing discharge processes and standards [64,65], and staged piloting within clinical services. Positive deviance approaches could be used to help identify and adapt practices from high-performing teams operating under similar constraints [66], ensuring strategies are feasible and contextually grounded. Leadership engagement and training to clarify role expectations are also critical to support adoption [64], while leveraging analytics capabilities can embed these improvements into routine service planning [67]. As this study demonstrates, unlocking iEMR’s potential could strengthen dietitians’ capability to plan and communicate nutrition-focused discharge care effectively, ultimately improving care continuity for older adults with malnutrition transitioning from hospital to home.

## 5. Strengths and Limitations

A key strength of this study is its multisite design, which captured diverse perspectives from frontline dietitians and senior leaders across varied hospital contexts, all using the same core iEMR system. The CFIR-guided analysis provided conceptual rigour in examining system-level, organisational and behavioural influences on iEMR use. Additionally, participation provided dietitians with an opportunity to reflect on their own discharge planning practices. Such reflective engagement is recognised as a characteristic that increases professionals’ buy-in and preparedness for future change initiatives [68]. However, findings may not be generalisable to non-digital hospital environments or regions using different EMR systems. They are also specific to hospital-based dietitians and may not directly apply to community dietitians, whose service contexts and roles differ. The absence of community-based providers’ perspectives limited insights into continuity from the receiving end of nutrition care transitions. Similarly, the lack of perspectives from medical team members limited understanding of interdisciplinary communication patterns related to nutrition care transitions. While participants’ social desirability bias cannot be entirely excluded, the specificity of reported workarounds and critiques suggests its influence was limited.

## 6. Future Research Directions

Future research should examine settings where iEMR is used effectively to support nutrition-focused discharge planning and care coordination, applying a positive deviance approach to identify transferable practices. Co-designing and piloting digital tools to help dietitians use iEMR more effectively represents a key next step. Studies should also explore how community providers caring for older adults with malnutrition experience receiving electronic nutrition care plans from hospitals. Additionally, examining perspectives across the full interdisciplinary team, including medical staff, could provide insights into optimising communication workflows and collaborative discharge planning processes. These efforts could inform targeted optimisation strategies that address both technical capabilities and organisational factors shaping nutrition care continuity.

## 7. Conclusions

“Making the system work” means equipping clinical dietitians with the digital tools, workflows and support they need to use iEMR effectively. However, as this study demonstrates, optimising iEMR alone is insufficient. Meaningful improvement requires addressing technical capabilities alongside organisational culture and policy frameworks that shape discharge practices. When these multi-level factors are appropriately aligned, these digital systems can better enable coordinated, nutrition-focused discharge care and improve continuity for older adults with malnutrition transitioning from hospital to home.

## Figures and Tables

**Table 1 healthcare-13-02227-t001:** Construct frequencies for CFIR 2.0 constructs applied during deductive coding of participant quotes about iEMR use for discharge planning and care coordination.

	CFIR DOMAIN					
CFIR CONSTRUCT	Inner Setting	Innovation	Outer Setting	Individual	Implementation Process	Total per Construct
Compatibility	86 (60)	-	-	-	-	146
Partnerships and Connections	-	-	98 (41)	-	-	139
Culture	76 (31)	-	-	-	-	107
Innovation Complexity	-	37 (64)	-	-	-	101
Innovation Relative Advantage	-	54 (41)	-	-	-	95
Communications	25 (55)	-	-	-	-	80
Reflecting and Evaluating—Innovation	-	-	-	-	48 (12)	60
Relational Connections	29 (18)	-	-	-	-	47
Innovation Adaptability	-	27 (17)	-	-	-	44
Roles—High-Level Leaders	-	-	-	30 (11)	-	41
Innovation Design	-	18 (20)	-	-	-	38
Access to Knowledge and Information	26 (6)	-	-	-	-	32
Characteristics—Capability	-	-	-	23 (6)	-	29
Characteristics—Motivation	-	-	-	15 (14)	-	29
Policies and Laws	-	-	11 (17)	-	-	28
Characteristics—Opportunity	-	-	6 (6)	-	-	12
Roles—Mid-Level Leaders	-	-	7 (0)	-	-	7
Relative Priority	1 (3)	-	-	-	-	4
Available Resources	0 (3)	-	-	-	-	3
Roles—Implementation Lead	-	-	-	3 (0)	-	3
IT Infrastructure	2 (0)	-	-	-	-	2
Tension for Change	0 (1)	-	-	-	-	1
**Total per Domain**	422	278	186	102	60	1048

Values represent the number of quotes coded to each CFIR 2.0 construct, grouped by domain. Row totals (values in the far right column) indicate the total number of times each construct was coded (primary and secondary), while column totals (bottom row) indicate the total number of codes within each domain. Values in parentheses indicate the number of times the construct was applied as a secondary (rather than primary) code. Only constructs coded at least once are shown; not all CFIR 2.0 constructs were observed in the dataset. IT = Information Technology.

**Table 2 healthcare-13-02227-t002:** Co-occurrence frequencies for CFIR 2.0 constructs applied to participant quotes about iEMR use for discharge planning and care coordination.

Construct A	Construct B	Co-Occurrence Count
Compatibility	Partnerships and Connections	32
Compatibility	Innovation Complexity	30
Partnerships and Connections	Innovation Relative Advantage	24
Compatibility	Culture	13
Culture	Partnerships and Connections	11
Culture	Communications	11
Compatibility	Communications	10
Relational Connections	Communications	10
Innovation Adaptability	Innovation Complexity	9
Partnerships and Connections	Communications	9
Compatibility	Innovation Relative Advantage	8
Innovation Relative Advantage	Reflecting and Evaluating	8
Partnerships and Connections	Innovation Complexity	8
Compatibility	Innovation Design	7
Innovation Design	Communications	7
Innovation Relative Advantage	Communications	7
Innovation Relative Advantage	Roles—High-Level Leaders	7
Reflecting and Evaluating	Innovation Design	7
Culture	Roles—High-Level Leaders	6

Pairs are listed in descending order of co-occurrence frequency. Construct pairs were derived by selecting the most frequently coded CFIR 2.0 constructs (from Table 1) and identifying all quotes in which any two of these constructs were applied together. Co-occurrence values represent the number of times two constructs were applied to the same quote, regardless of coding order (i.e., whether as primary or secondary codes). Thematic analysis was conducted on all pairs until no new themes were identified, which occurred during analysis of the Culture and Roles—High-Level Leaders pairing.

**Table 3 healthcare-13-02227-t003:** Summary of overarching themes and subthemes describing dietitians’ use of iEMR for discharge planning and care coordination.

Theme	Theme Name	Theme Description	Subtheme	Subtheme Name	Subtheme Description
1	Technical System Limitations and Policy Constraints	System-level design, usability, and interoperability issues, and external policy constraints undermined effective discharge planning and coordination	1.1	Technical fragmentation, poor interoperability and design limitations compromise care continuity	Fragmented workflows, poor integration and usability barriers disrupted discharge processes and trust in digital documentation
			1.2	Policy and program eligibility restrictions limit service access and care coordination	External funding and eligibility rules constrained post-discharge dietetic care, often overriding clinical judgement.
2	Adaptations and Workarounds in Daily Practice	Dietitians developed practical workarounds and informal communication strategies to navigate system barriers and ensure continuity of care	2.1	Trust concerns drive reliance on parallel communication methods	Persistent trust concerns led dietitians to duplicate digital documentation with verbal, phone or email confirmations.
			2.2	Documentation and communication workarounds emerge to overcome system functionality limitations	Informal adaptations supported workflow efficiency but introduced variability and undermined standardisation efforts.
3	Organisational Culture and Norms	Social, cultural and professional norms shaped discharge planning practices beyond technical constraints	3.1	Practice variation is shaped by inconsistent cultural expectations and norms	Divergent documentation practices were reinforced by inconsistent expectations, leadership support and local team culture
			3.2	Dietitians’ evolving role and visibility in discharge planning	Dietitians’ evolving role identity and perceived marginalisation from discharge decision-making processes reflected broader interdisciplinary hierarchies.
			3.3	Relational coordination often supersedes structured workflows	Informal interpersonal networks often replaced or supplemented structured discharge coordination processes.
4	Optimising iEMR: Future Directions and System Potential	Participants envisioned opportunities to better integrate, standardise and leverage iEMR to enhance discharge planning and service delivery	4.1	Confidence in system value varies by clinical context and user experience	Confidence in iEMR use varied by context; praised for coordination within the same health service but presenting greater challenges at external service boundaries.
			4.2	Analytical capabilities hold potential for transforming service delivery	Dietitians identified aspirations for structured data use, dashboards and outcome tracking to drive future service improvements.
			4.3	Integrated and standardised workflows promise more efficient discharge planning	Participants advocated for unified, streamlined discharge planning workflows to enhance usability, quality and statewide consistency.

Grey shading indicates rows that present both the overarching theme and its first subtheme. Themes and subthemes were derived from construct-level and co-occurrence thematic analysis of dietitians’ experiences with iEMR use for discharge planning and care coordination.

## Data Availability

The qualitative data generated and analysed during this study are not publicly available due to confidentiality and ethical requirements. De-identified excerpts supporting the findings are included in Appendix A. Further information may be available from the corresponding author upon reasonable request, subject to ethical approval.

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
