# Peer review of "‘Making the System Work’: A Multi-Site Qualitative Study of Dietitians’ Use of iEMR to Support Nutrition Care Transitions for Older Adults with Malnutrition"

_healthcare, 2025, doi:10.3390/healthcare13172227_

Round 1

Reviewer 1 Report

Comments and Suggestions for Authors

Please find attached our comments on the draft. This is an interesting study; however, it still needs improvement in how the author writes the introduction section and the method.

Reviewer 2 Report

Comments and Suggestions for Authors

This qualitative study provides an overview of the use of the integrated electronic medical record (iEMR) by 16 dietitians in Australia. The authors identified four themes and ten subthemes during the coding process. Overall, the article is well-written.

Major comments:

  1. In qualitative studies, it is a common practice to include at least two coders. Please justify the use of only one coder for this study.
  2. Please include the information on whether the participants got compensation for their time.
  3. Please include the reflexivity practice details during data coding and analysis.
  4. Discussion: Please include examples from other qualitative and/or quantitative studies on the importance of iEMR to assist people during care transitions.
  5. Discussion: I suggest exploring the innovation suggestions further and how to implement them in clinical practice, as this will improve the impact of this study.

Minor comments:

  1. Lines 163-164: Please include the citation of the approach performed.

Round 2

Reviewer 1 Report

Comments and Suggestions for Authors

We accepted the revision from the authors. This version can continue to the further steps.

Reviewer 2 Report

Comments and Suggestions for Authors

The authors have satisfactorily addressed all the comments.